# Effects of simulated Secondary Organic Aerosol Water on PM$_1$ levels and composition over US

Stylianos Kakavas[1,2], Spyros N. Pandis[1,2] and Athanasios Nenes[1,3]

[1]Institute of Chemical Engineering Sciences, Foundation for Research and Technology Hellas, Patras, Greece

[2]Department of Chemical Engineering, University of Patras, Patras, Greece

[3]School of Architecture, Civil and Environmental Engineering, École Polytechnique Fédérale de Lausanne (EPFL), Switzerland

*Correspondence to*: Athanasios Nenes (athanasios.nenes@epfl.ch) and Spyros N. Pandis (spyros@chemeng.upatras.gr).

**Abstract.** Water is a key component of atmospheric aerosol, affecting many aerosol processes including gas/particle partitioning of semi-volatile compounds. Water related to secondary organic aerosol (SOAW) is often neglected in atmospheric chemical transport models and is not considered in gas-to-particle partitioning calculations for inorganic species. We use a new inorganic aerosol thermodynamics model, ISORROPIA-lite, which considers the effects of SOAW, to perform chemical transport model simulations for a year over the continental United States to quantify its effects on aerosol mass concentration and composition. SOAW can increase average fine aerosol water levels up to a factor of two when secondary organic aerosol (SOA) is a major PM$_1$ component. This is often the case in the south-eastern U.S where SOA concentrations are higher. Although the annual average impact of this added water on total dry PM$_1$ concentrations due to increased partitioning of nitrate and ammonium is small (up to 0.1 μg m$^{-3}$), total dry PM$_1$ increases of up to 2 μg m$^{-3}$ (with nitrate levels increases up to 200%) can occur when RH levels and PM$_1$ concentrations are high.

## 1. Introduction

Atmospheric particulate matter with aerodynamic diameter smaller than 1 μm (PM$_1$) has adverse effects on public health, climate and ecosystem productivity (Pye et al., 2020; Baker et al., 2021; Guo et al., 2021). PM$_1$ is composed of thousands of organic compounds, black carbon (BC), and inorganic components such as sulfate (SO$_4^{2-}$),

nitrate ($NO_3^-$), ammonium ($NH_4^+$) and chloride ($Cl^-$) (Seinfeld and Pandis, 2006). Ambient aerosol is mostly composed of water which is determined by the chemical equilibrium of water vapor with the aerosol constituents (Liao and Seinfeld, 2005; Carlton and Turpin, 2013; Bian et al., 2014; Guo et al., 2015; Bougiatioti et al., 2016; Nguyen et al., 2016; Guo et al., 2017; Deetz et al., 2018; Kuang et al., 2018; Song et al., 2018; Wu et al., 2018; Pye et al., 2020; Gopinath et al., 2022). Aerosol liquid water directly affects the PM sensitivity and dry deposition rates, with direct implications for emissions control policy (Nenes et al., 2020; Nenes et al., 2021; Sun et al., 2021).

The hygroscopicity parameter ($\kappa$), which expresses the ability of a PM component to absorb water, is an effective approach for the parameterization of the water uptake of atmospheric PM that is a mixture of organic and inorganic species (Petters and Kreidenweis, 2007). Although organic aerosol (OA) is less hygroscopic than inorganic salts, it can still contribute significantly to the total aerosol water (Guo et al., 2015; Bougiatioti et al., 2016; Jathar et al., 2016; Li et al., 2019) or can even become the dominant contributor at lower ambient relative humidity (Jin et al., 2020). Previous studies have demonstrated that secondary organic aerosol (SOA) is a lot more hygroscopic ($0.1 \leq \kappa \leq 0.3$) than primary organic aerosol (POA) ($\kappa \leq 0.01$) and is mainly responsible for the corresponding OA water (Petters et al., 2006; Koehler et al., 2009; Chang et al., 2010; Jathar et al., 2016; Kuang et al., 2020; Li et al., 2020).

SOAW can enhance secondary inorganic aerosol concentrations assisting in their partitioning in the particulate phase to satisfy equilibrium. However, such effects are not considered in thermodynamic modules used for the simulation of gas-to-particle partitioning of inorganic species in chemical transport models. Evidence exists however that fine aerosol nitrate and ammonium concentrations can increase in areas with high organic aerosol and RH levels (Kakavas et al., 2022). The importance of these SOAW impacts on secondary aerosol formations has not been systematically studied and is the focus of this work.

We use a new aerosol thermodynamics model, ISORROPIA-lite (Kakavas et al., 2022), to simulate SOAW effects on the partitioning of the inorganic components, for a year over the continental United States. The model performance has been evaluated for fine PM and its components for the examined period by Skyllakou et al. (2021). It is considered good for the total $PM_{2.5}$ concentration and average for most of the components. The aim of our work is to quantify the SOAW contribution to the total

fine PM water and to study its effects on inorganic aerosol thermodynamics and total
dry fine PM levels and composition.

**2. Methods**
**2.1 ISORROPIA-lite**
ISORROPIA-lite is a lean and accelerated version of the widely used ISORROPIA-II
(Fountoukis and Nenes, 2007) aerosol thermodynamics model and it focuses on the
simulation of the composition of the inorganic fraction of the atmospheric aerosol that
is in equilibrium with the gas phase. It assumes that the aerosol exists only in the
metastable state at low RH and the activity coefficients of ionic pairs are always
obtained from precalculated look-up tables. It estimates aerosol water associated with
each one of the aerosol components. Furthermore, ISORROPIA-lite has an important
additional feature compared to ISORROPIA-II, as it considers the effects of SOAW on
inorganic aerosol thermodynamics. The resulting increase of the total water mass drives
more of the water-soluble gaseous species to the particle phase to satisfy equilibrium.
SOAW, $W_{SOA}$, in ISORROPIA-lite is calculated using the well-established $\kappa$-Kohler
theory of Petters and Kreidenweis (2007):
$$W_{SOA} = \frac{\rho_w}{\rho_{SOA}} \frac{C_{SOA}\kappa}{\left(\dfrac{1}{RH}-1\right)} \tag{1}$$

where $\rho_w$ is the density of water, $\rho_{SOA}$ the SOA density, $C_{SOA}$ the SOA concentration, $\kappa$
the SOA hygroscopicity parameter and RH the relative humidity in the 0−1 scale. More
details about the ISORROPIA-lite can be found in Kakavas et al. (2022).

**2.2 PMCAMx description and application**
PMCAMx (Karydis et al., 2010; Tsimpidi et al., 2010) is a three dimensional chemical
transport model based on CAMx (Environ, 2006), which simulates horizontal and
vertical advection and dispersion, dry and wet deposition, as well as aqueous, gas, and
aerosol chemistry. The mechanism used in this work for gas-phase chemistry
simulations is the Carbon Bond 05 (CB5) (Yarwood et al., 2005) and includes 190
reactions of 79 gas species. To describe the aerosol size and composition distribution
10-size sections (from 40 nm to 40 μm) are used assuming that all particles in each size
bin have the same composition. Therefore, PMCAMx predicts $PM_x$ concentrations
where $x$ can be among other choices 1, 2.5 and 10 μm. Equilibrium is always assumed
between the bulk aerosol and gas phases. The partitioning of semi-volatile inorganic
species between the gas and particulate phases is simulated by ISORROPIA-lite.
Weighting factors based on the effective surface area of each size bin are used to
distribute to the various size bins the mass transferred between the two phases in each
time step (Pandis et al., 1993). For the simulation of organic aerosols, the volatility
basis set (VBS) approach (Donahue et al., 2006) is used. POA is simulated using eight
volatility bins (from $10^{-1}$ to $10^{6}\,\mu g\,m^{-3}$) at 298 K, while for SOA four volatility bins (1,
10, $10^{2}$, $10^{3}\,\mu g\,m^{-3}$) at 298 K are used (Murphy and Pandis, 2009). An organic aerosol
phase is assumed with its components forming a pseudo-ideal solution and being in
equilibrium with the gas phase (Strader et al., 1998). The influence of water on the
partitioning of the organic components of the particulate matter between the gas and
aerosol phases is assumed to be negligible in this version of PMCAMx (Koo et al.,
2003). The low volatility organic compounds (LVOCs) and the extremely LVOCs are
implicitly included in the lowest volatility bin of this version of the VBS used in
PMCAMx. These compounds are always in the particulate phase in these simulations
and therefore the addition of lower bins increases the computational cost without
changing the predicted organic aerosol concentration. For the major point sources, the
NO$_x$ plumes are simulated using the Plume-in-Grid (PiG) approach (Karamchandani et
al., 2011; Zakoura and Pandis, 2019). The variable size resolution model (VSRM) of
Fahey and Pandis (2001) is used for the simulation of aqueous-phase. The model is
based on the chemical mechanism of Pandis and Seinfeld (1989) with the addition of
$Ca^{2+}$ to the list of particle components as well as $H_2SO_4$ in the gas phase (Fahey and
Pandis, 2001).
We applied PMCAMx over the continental United States during 2010. The
modeling domain includes northern Mexico and southern Canada and covers a 4752 $\times$
2952 km$^2$ region (Figure S1). The model grid consists of 10,824 cells with horizontal
dimensions of 36 $\times$ 36 km. The meteorological inputs were provided by the Weather
Research Forecasting model (WRF v3.6.1) using a horizontal resolution of 12 $\times$ 12 km.
RH levels above 95% were rare therefore there was no need for screening of the few
high RH values outside clouds. The gaseous and primary particle emissions were
developed by Xing et al. (2013). More details about the meteorological inputs and the
emissions can be found in Skyllakou et al. (2021).
To quantify the SOAW effects on inorganic aerosol thermodynamics three
simulations were performed. The first was a simulation neglecting SOAW and
including only inorganic aerosol water. Two additional simulations were performed:
one where $\kappa$ of SOA was assumed to be equal to 0.1 and one with $\kappa$=0.2 (Kuang et al.,
2020) to examine how SOA hygroscopicity affects total fine aerosol water content and
PM levels and composition. Even if higher values of the hygroscopicity parameter (e.g.,
$\kappa$=0.3) are possible (Kuang et al., 2020) these represent rather extreme cases for the
simulation of the average effects over the US. Therefore, the two simulations used in
this study provide a good estimate of the corresponding uncertainty. Previous studies
have estimated secondary organic aerosol density values of 1−1.4 g cm$^{-3}$ (Turpin and
Lim, 2001; Kostenidou et al., 2007). A SOA density of 1 g cm$^{-3}$ was assumed in the
simulations. The SOA exists mostly in submicrometer particles so our subsequent study
focuses on PM$_1$.

**3. Results**
**3.1 Effects of SOAW on PM$_1$ water levels**
The annual average PM$_1$ water ground-level concentrations neglecting SOAW are
shown in Figure 1. Higher PM$_1$ water concentrations from 8 to 18 μg m$^{-3}$ are predicted
in the north-eastern part of the US due to the higher inorganic PM$_1$ concentrations
(Figure 2) and RH levels in that area. When SOAW is present in the simulations, total
PM$_1$ water levels increase everywhere with higher fractional increases in the south-
eastern US (up to 50% when $\kappa$=0.1 and up to 100% when $\kappa$=0.2 in Alabama and north-
western Mexico) due to higher SOA levels (Figure S2). In the north-eastern US, lower
fractional increases are predicted (10−15% when $\kappa$=0.1 and 20−30% when $\kappa$=0.2). In
general, assuming a $\kappa$ of SOA equal to 0.2 instead of 0.1 increases the corresponding
amount of SOAW by about a factor of two. Figure 1 shows the distributions of
fractional increase change in the annual PM$_1$ water levels at ground level from SOAW.
Total PM$_1$ water average concentrations increase from 20 to 30% in about 60% of the
modeling domain when $\kappa$=0.1. For $\kappa$=0.2, the corresponding increase is from 40 to 60%.

Predicted SOA levels are higher during summertime (Figure S2) since the

emissions and oxidation rates of volatile organic compounds (VOCs) are higher (Zhang
et al., 2013; Freney et al., 2014; Skyllakou et al., 2014; Fountoukis et al., 2016).
However, even during wintertime fresh biomass burning emissions exposed to NO$_2$ and
O$_3$ can form significant amounts of SOA in periods with low OH levels (Kodros et al.,
2020). Higher total PM$_1$ water concentrations are predicted during winter (Figure 3)
since the RH levels and inorganic fine aerosol concentrations are higher; especially

nitrate and chloride which increasingly partition to the aerosol phase as temperature decreases (Guo et al., 2017). However, $PM_1$ chloride concentrations are low (less than $0.1\ \mu g\ m^{-3}$) with higher concentrations in Kansas because of biomass burning episodes. Higher fractional increases in fine aerosol water levels (up to 5 times) due to SOAW are predicted during summer in the south-eastern part of US where SOA concentrations are higher. This corresponds to increases to average fine aerosol water concentrations up to $8\ \mu g\ m^{-3}$.

Ammonium nitrate and ammonium sulfate are the inorganic salts that contribute the most to the total $PM_1$ water levels (Figure S3). SOAW also contributes significantly to the total $PM_1$ water levels especially in the south-eastern US (about 30 and 50% of total $PM_1$ water when $\kappa=0.1$ and $\kappa=0.2$ respectively), when the mass fraction of SOA in dry $PM_1$ exceeds 30%.

**3.2 Effects of SOAW on total dry $PM_1$ levels**

Higher dry $PM_1$ concentrations are predicted for the eastern part of the US (up to 15 $\mu g$ $m^{-3}$) in the base case (Figure 4). These dry $PM_1$ levels increase slightly up to 0.6% and 1.2% due to SOAW when $\kappa=0.1$ and $\kappa=0.2$ for SOA is assumed. The highest annual average fractional increase in total dry $PM_1$ levels is predicted in California (1% when $\kappa=0.1$ and 2% when $\kappa=0.2$). The probability density (Figure 4) indicates that in about 60% of the modeling domain total dry fine aerosol concentrations increase up to 0.3% when $\kappa=0.1$. For $\kappa=0.2$, the corresponding increase is from 0.4 to 2%. The areas of the highest $PM_1$ increase correspond to regions where aerosol pH tends to be relatively high (Pye et al., 2020). In these areas, nitric acid and ammonia can condense and increase aerosol mass because of the increase in water from the SOA. Because of this partitioning change, the predicted gas-phase concentrations of semi-volatile inorganic components decreased on average when SOAW was considered (Figure S4). SOAW had a negligible absolute impact on the small fine chloride concentrations in this period (Figure 2). However, in periods during which chloride salts and SOA contribute significantly to the total dry (e.g. during intense biomass burning periods), fine chloride concentrations could also change (Metzger et al., 2006; Fountoukis et al., 2009; Gunthe et al., 2021).

Skyllakou et al. (2021) found that PMCAMx had a small fractional bias (5%) and a fractional error (25%) for the annual average $PM_{2.5}$ concentrations of 1067 measurements stations in the U.S. The performance of PMCAMx regarding annual

average OA is considered good in these simulations with a fractional bias of 5% and a
fractional error of 26% in the 306 stations in the US. For daily average concentrations
the performance is also quite encouraging with a fractional bias of 15% and a fractional
error of 56%. Given that the effect of the extension of the model on the total fine PM
mass is small (of the order of 1%), this does not result in any noticeable change in its
already very good performance for dry fine PM. Therefore, the major change in the
model predictions is on the aerosol water concentrations.

**3.3 Effects of SOAW on PM$_1$ components**
The annual average results indicate that SOAW mainly affects fine aerosol water levels.
To better analyze the effects of SOAW we focus on the temporal evolution of the
predicted levels of PM$_1$ components in four sites (Figure S1) with different
characteristics (Table S1). We have chosen one city from the West, one from the South,
one from Southeast and one from the Northeast. They are all in different environments
with different major sources and climatological conditions. The presence of SOAW
increased PM$_1$ water concentrations in all sites from 1% to almost an order of
magnitude (Figure 5). However, these fractional increases most of the time correspond
to PM$_1$ water concentration increases of a few μg m$^{-3}$ (Figure S5) because they occur
under low RH levels. During higher RH periods (80 to 100%), the PM$_1$ water levels are
predicted to increase up to 100 μg m$^{-3}$ (e.g. in Toronto).

Total dry PM$_1$ concentrations during most of the simulated period increase on

average less than 1% in all sites (Figure 5) due to SOAW. There are periods, however,
with higher fractional increases (up to 10%) and even small decreases (up to 5%) in
total dry fine aerosol levels in the examined sites. The decreases can be explained
because SOAW increases the size of particles and therefore their dry deposition rate
(Nenes et al., 2020). Depending on SOA hygroscopicity, increases up to 1.5 μg m$^{-3}$ for
nitrate and 0.5 μg m$^{-3}$ for ammonium are predicted (Figure S5). Fine nitrate increases
of 10% were more frequent in the examined sites; however higher increases up to 200%
are predicted during the simulated period (Figure 6). As expected, higher increases can
occur more often with higher assumed SOA hygroscopicity.

**4. Discussion**
Aerosol liquid water has a profound impact on aerosol processes, chemical composition
and their impacts. By including the effects of organic water on inorganics

thermodynamic equilibrium we show that SOAW can substantially increase aerosol water levels, on an average up to 60% over the majority of the domain. As a consequence, total dry $PM_1$ levels can also increase but the changes are small (up to 2% on an annual average basis). Locally these effects can be much more significant during periods of high RH and SOA levels (fine nitrate fractional increases can be as high as 200%).

The effects vary with season. During summer, the RH is lower and SOA levels are higher leading to higher fractional increases in aerosol water (Figure 3) but lower absolute mass changes. During summer the fractional increases in total dry fine aerosol concentrations are lower than in wintertime (Figure S6). Responsible for the total dry fine aerosol concentration increases are nitrate and ammonium (Figure 2). These compounds partition together (as deliquesced ammonium nitrate) to the particulate phase to satisfy equilibrium due to the additional water mass of SOA.

The increases in total dry $PM_1$ and fine aerosol water levels depend on SOA concentrations, hygroscopicity value, RH levels and the particle phase fractions of inorganic species. The SOAW effect on aerosol water is approximately proportional to the assumed hygroscopicity parameter $\kappa$. Given that our work investigates the potential significance of this effect we have chosen to provide the results of two simulations one with relatively low and relatively high hygroscopicity of SOA. A more detailed treatment of the hygroscopicity parameter (e.g., assigning a different value to each OA component) will be a topic of future work.

The present work, thoroughly analyzes organic water uptake impacts over one simulated year (not just one month as done in Kakavas et al., 2022) and in quite a different geographical area (US here versus Europe in Kakavas et al., 2022). There are significant differences, but also similarities in the predicted changes and effects of SOA water. Both studies indicate that SOAW can contribute highly to the total $PM_1$ water and increase particulate nitrate concentrations especially in areas with high total nitrate concentrations. Pilinis et al. (1995) have argued that the single most important parameter in determining direct aerosol forcing is RH, and the most important process is the increase of the aerosol mass as a result of water uptake. They estimated that on average an increase of the RH from 40 to 80% for a global mean aerosol more than doubles the corresponding radiative forcing. As a result the inclusion of SOAW in future studies is highly recommended. ISORROPIA-lite provides a simple and computationally effective approach for the simulation of this SOAW.

*Code and Data Availability.* The model code and data used in this study are available from the authors upon request (spyros@chemeng.upatras.gr and athanasios.nenes@epfl.ch).

*Author Contributions.* SK incorporated ISORROPIA-lite in PMCAMx, carried out the simulations, analyzed the results and wrote the manuscript. SN and AN conceived and led the study and helped in the writing of the manuscript. All authors contributed to the reviewer responses and manuscript revisions.

*Competing Interests.* The authors declare no competing financial interest.

*Acknowledgements.* This work was supported by the project FORCeS funded from the European Union's Horizon 2020 research and innovation programme under grant agreement No 821205, and project PyroTRACH (ERC-2016-COG) funded from H2020-EU.1.1. - Excellent Science - European Research Council (ERC), project ID 726165.

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

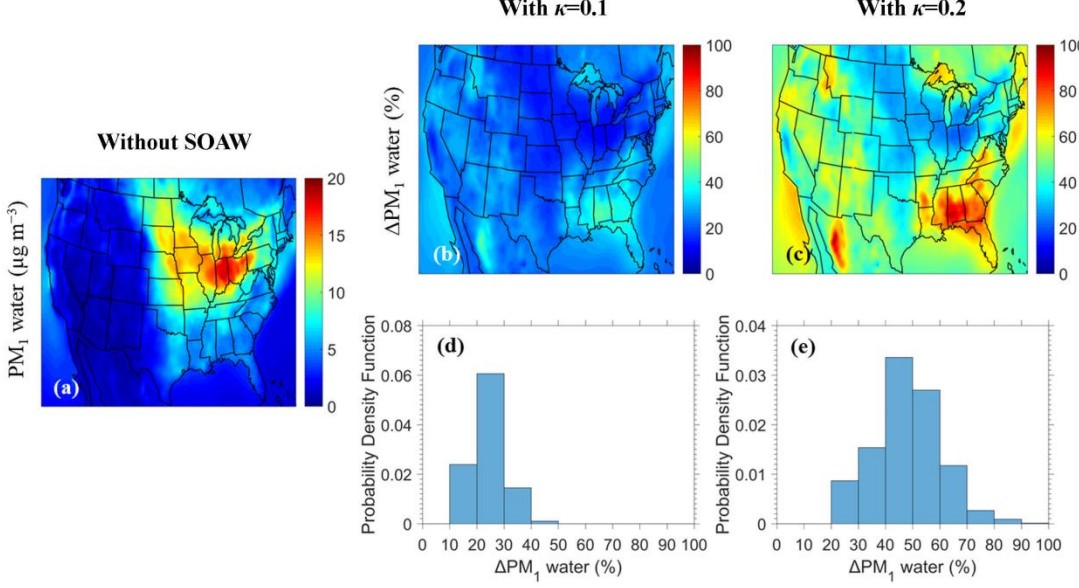



**Figure 1.** Maps of: **(a)** annual average $PM_1$ water ground-level concentrations neglecting SOAW, **(b)** annual average fractional increase of $PM_1$ water when SOAW is present in the simulations with $\kappa$=0.1 and, **(c)** with $\kappa$=0.2 during 2010. The probability density as a function of fractional increase in the annual $PM_1$ water concentrations due to SOAW when: **(d)** $\kappa$=0.1 and **(e)** $\kappa$=0.2 is shown.


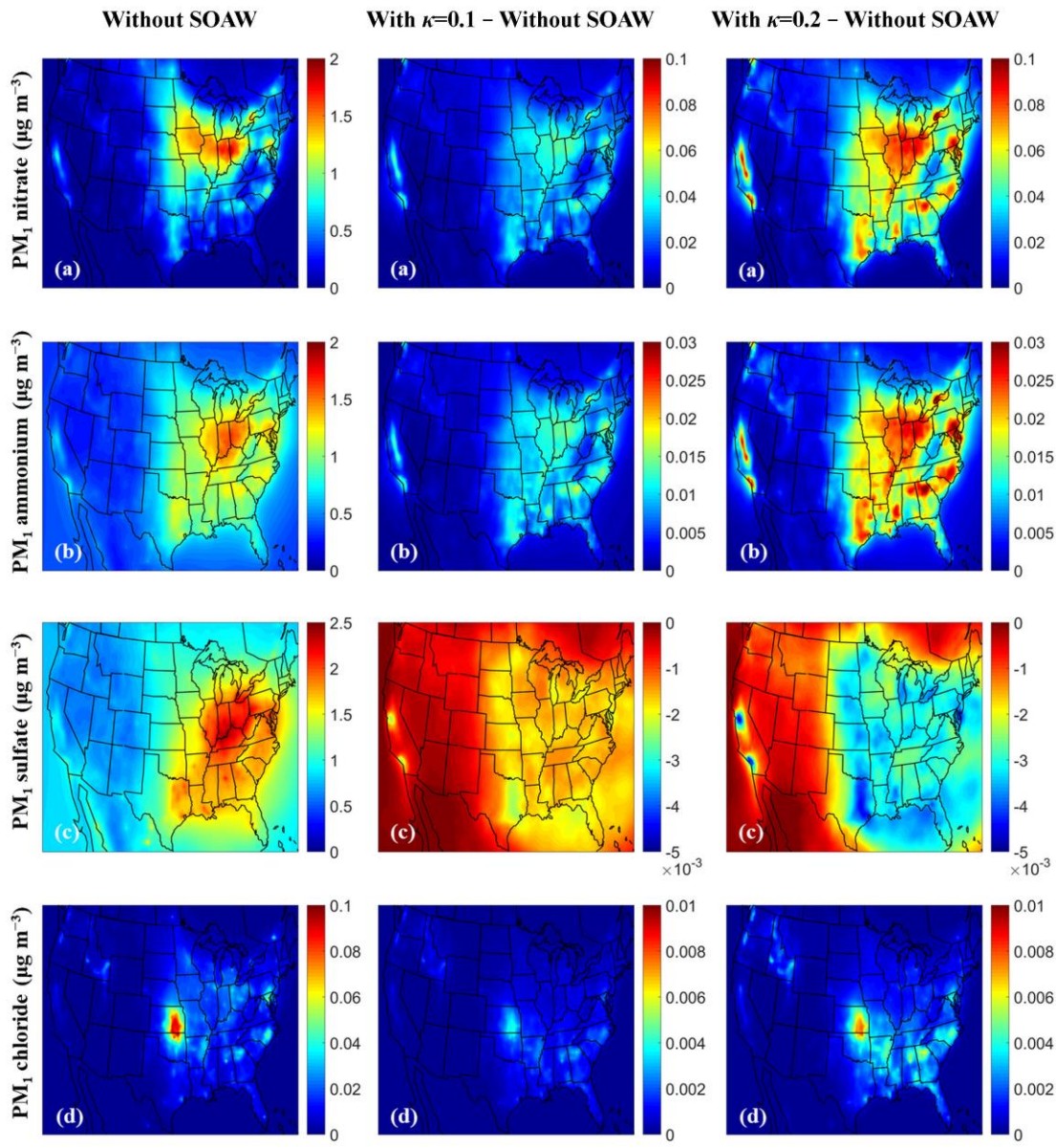



**Figure 2.** Annual average ground-level concentrations (in μg m$^{-3}$) of PM$_1$: **(a)** nitrate,
**(b)** ammonium, **(c)** sulfate, and **(d)** chloride neglecting SOAW and the annual
concentration changes when SOAW is present in the simulations with $\kappa$=0.1 and $\kappa$=0.2.
A positive change corresponds to an increase. A negative change corresponds to a
decrease.

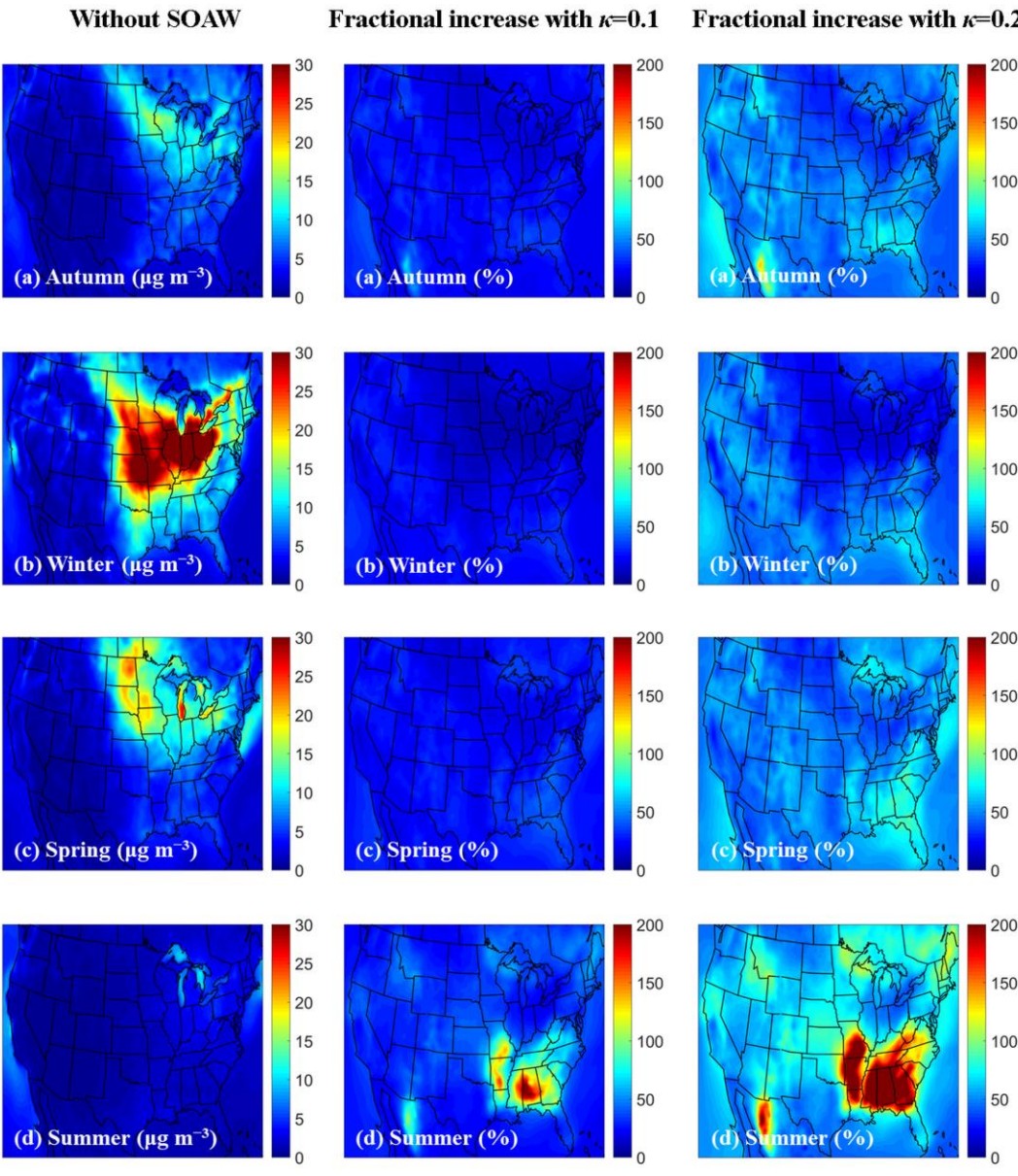

**Figure 3.** Average ground-level concentrations of PM$_1$ water neglecting SOAW (in μg m$^{-3}$) and the fractional increase when SOAW is present in the simulations with $\kappa$=0.1 and $\kappa$=0.2 during: **(a)** autumn (SON), **(b)** winter (DJF), **(c)** spring (MAM), and **(d)** summer (JJA) of 2010.

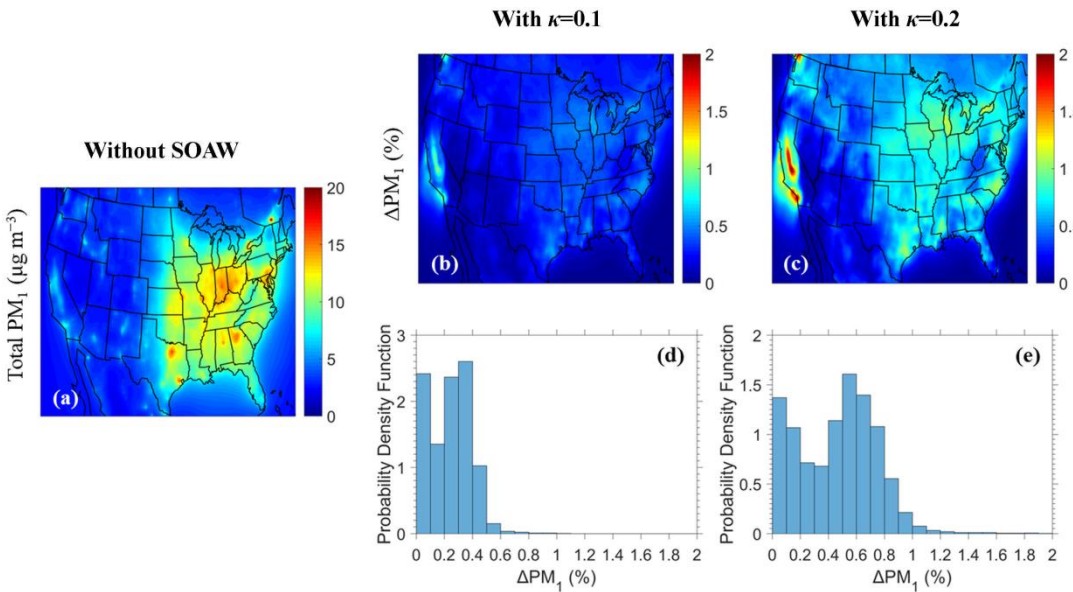

**Figure 4.** Maps of: **(a)** annual average total dry $PM_1$ ground-level concentrations neglecting SOAW, **(b)** annual average fractional increase of total dry $PM_1$ when SOAW is present in the simulations with $\kappa$=0.1 and, **(c)** with $\kappa$=0.2 during 2010. The probability density as a function of fractional increase in the annual total dry $PM_1$ concentrations due to SOAW when: **(d)** $\kappa$=0.1 and **(e)** $\kappa$=0.2 is shown.

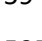



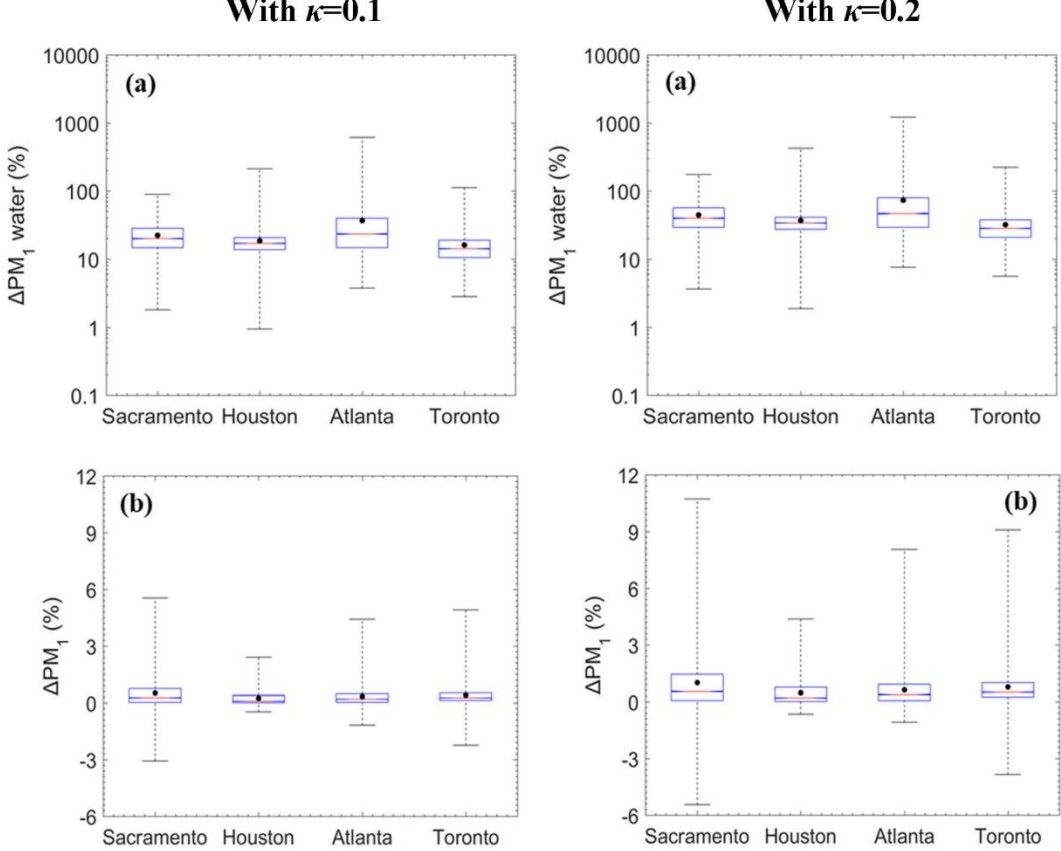



**Figure 5.** Box plots for fractional change in the hourly: **(a)** $PM_1$ water and **(b)** total dry $PM_1$ due to SOAW when $\kappa=0.1$ and $\kappa=0.2$ for Sacramento, California; Houston, Texas; Atlanta, Georgia; and Toronto, Canada during 2010. The red line represents the median, the black dot is the mean value, the upper box line is the upper quartile (75%) and the lower box line is the lower quartile (25%) of the distribution. A negative change corresponds to a decrease.


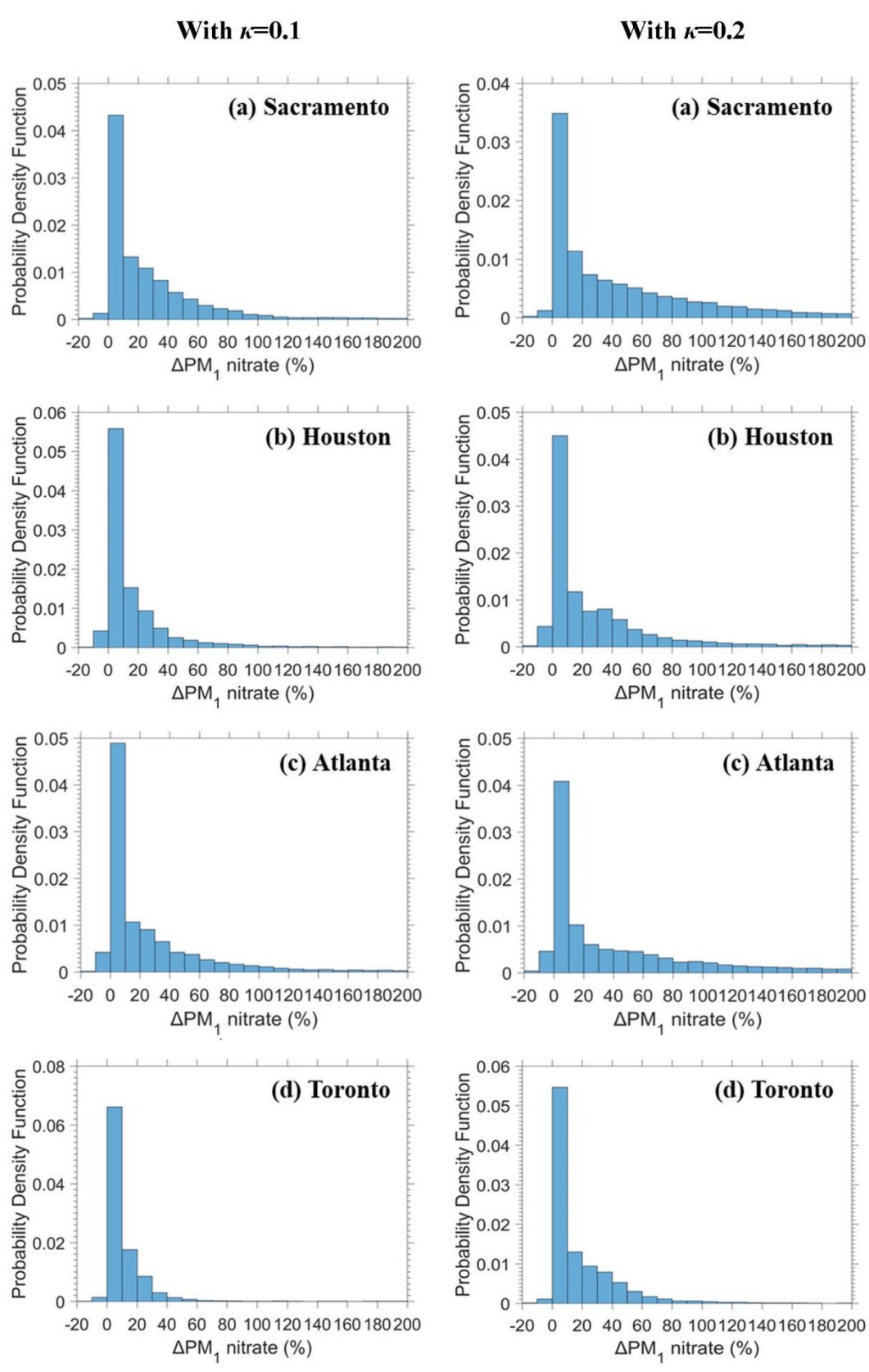


**Figure 6.** The probability density as a function of fractional increase in the hourly $PM_1$ nitrate due to SOAW when $\kappa$=0.1 and $\kappa$=0.2 for: **(a)** Sacramento, California; **(b)** Houston, Texas; **(c)** Atlanta, Georgia; and **(d)** Toronto, Canada during 2010.