# Peer review of "Effects of simulated secondary organic aerosol water on fine PM levels and composition over US"

_Atmospheric Chemistry and Physics, 2022_

## Author Comment (AC1)

**Response to the Comments of Reviewer 1**

*(1) This manuscript describes air quality model simulations with PMCAMx of North America with focus on the U.S. The model simulations include ISORROPIA-lite to simulate aerosol liquid water not only from inorganic PM constituents, but with water uptake contributed from secondary organic aerosol (SOA) constituents. The authors investigate the amount and relative change in aerosol water, dry mass from nitrate, HCl, HNO₃, and ammonium due to this process. The authors find ubiquitous increase in predicted wet and dry PM₁ mass concentrations. The authors identify an important and interesting topic regarding interactions among water uptake, organic aerosol species formed in situ and the impacts on particle-phase chemical composition.*

We thank the reviewer for pointing out the importance of this study. Our responses and corresponding changes to the manuscript (in regular font) follow each comment of the reviewer (in italics).

*(2) There is no connection of model predictions to measurements and it is difficult to understand if the changes represent improved predictive skill. The authors generally provide little to no support for employed values (e.g., kappa, density) or the city selection. If statistical tests were performed, for example, to determine that the 1% difference in PM₁ dry mass is statistically significant – they are not discussed. Would such change be sufficient to be detected in an observational network or during a field campaign? I cannot support publication of this manuscript in its present form. Provided the comments below are addressed the manuscript may be publishable.*

We have done our best to address each comment of the reviewer and to improve the manuscript accordingly. Unfortunately, the most important predicted variable for the current simulations, the aerosol liquid water, is a challenge to measure – and for that reason no particle liquid water content measurements are thus available for the model evaluation. A detailed model performance evaluation for the rest of the $PM_{2.5}$ components has been provided by Skyllakou et al. (2021). Given the small changes in the average concentrations of the major $PM_{2.5}$ components when the secondary organic aerosol water is included in the simulation the performance of the model remained practically the same. It is considered good for the total $PM_{2.5}$ concentration and average for most of the components. This information has been added to the revised paper.

We have added explanations and the corresponding references for the employed values for the hygroscopicity parameter, density, etc. and also about our city selection. We now clarify in the paper that a 1% change in $PM_1$ is not important for all practical purposes. We now clarify that the change is very small and we just mention the exact number in parenthesis to avoid confusion about what we consider very small. That said, there are still cases where the changes are important, and we still keep those points in the revision – always noting conditions for which they occur.

**Detailed Comments**

*(4) This a 3-D modeling study, and authors make no connection to field observations. The title should reflect that. For example, "Effect of simulated …."*

Good point. We have added information about the model performance for total $PM_{2.5}$ and the concentrations of the major $PM_{2.5}$ components during the simulation. We have made the proposed change in the title of the paper.

*(5) The authors motivate their work with discussion of $PM_{2.5}$, and classify all of their results in terms of $PM_1$. Why the disconnect? Further, how was $PM_1$ calculated from model output? This is not described in main text or supplemental information.*

PMCAMx simulates the aerosol size/composition distribution using a sectional approach so it predicts $PM_x$ where *x* can be among other choices 1, 2.5 and 10 μm. This is now explained in the model description. The disconnect is mainly due to the regulation of $PM_{2.5}$ and not $PM_1$. We agree with the point of the reviewer though and we have focused on $PM_1$ in our discussion in the revised paper.

*(6) Line 37: "Potassium levels can be significant … biomass burning" This sentence seems a little out of place, especially given the Cl discussion regarding biomass burning later in the manuscript (lines 171-173). Also, is the review paper by Pye et al, the best reference for this point?*

Good point. We have deleted this sentence from this point in the manuscript together with the corresponding reference.

*(7) The HCl hotspot in KS should be addressed. In the text, chlorine species are discussed primarily in relation to their presence due to biomass burning, which (I don't think) is happening in the KS hotspot.*

A brief discussion of the source of this rather low chloride concentration ($\sim 0.1 \ \mu g \ m^{-3}$) in this area has been added to the paper.

*(8) Line 50: Does "a lot more hygroscopic" have a quantitative meaning?*

Indeed so, as on a per mass basis SOA can uptake multiple times (depending on source and ageing, a factor of 2-3) water compared to primary OA. We have rephrased the corresponding sentence to reflect this.

*(9) Line 117 and Line 119: can justification be provided for the Kappa and SOA density values? Can the authors defend the use of the kappa values in the context of a regional simulation or evaluation focused on discussion of urban areas? In the simulations here, aromatic SOA has the same hygroscopic properties and density as 'aged' SOA? Why not pick a higher and lower bounds-0.3 to 0.05? Or better yet, why not apply k values based on chemical information of SOA species since the authors have that information from the model? Any which way, some reasoning for the chosen kappa values is needed.*

We have added a brief discussion of the hygroscopicity parameter values in the literature and provided justification of our choices in the revised paper. Given that our paper investigates the potential significance of this effect we have chosen to provide the results of two simulations one with relatively low and relatively high hygroscopicity of SOA, which bound the range of the

corresponding effects of the SOA water on particle water, mass and impacts on semi-volatile inorganic species partitioning. A more detailed treatment of the hygroscopicity parameter (e.g., assigning a different value to each OA component) is of course possible – and is left to be the topic of future work. We have also added a discussion of this point in the end of the paper.

*(10) Why were the particular cities selected? Why are they introduced in the end?*

Our intention is to look into more detail into the behavior of SOA water and its effects in selected locations in the domain. We have chosen one city from the West, one from the South, one from the Southeast and one from the Northeast. They are all in different environments with different major sources and climatological conditions. We explain our reasoning in the revised paper.

*(11) It is difficult to accurately measure RH above ~95%. Did the authors screen out any RH values when evaluating water mass predictions?*

This is a relevant point for the analysis of measurements. Given that the RH is predicted in this case and it does not suffer from the corresponding experimental challenges there was no need for screening of the few values above 95%. In fact, it points to the potential need for measurements at ultra-high RH – which is quite possible (e.g., https://acp.copernicus.org/articles/10/1329/2010/) but seldomly carried out by research groups. We will clarify these points in the revision.

*(12) The authors state "The model performance has been evaluated for fine PM and its components for the examined period by Skyllakou et al. (2021)." What did they find? For example, in these simulations there is a universal increase in $PM_1$ mass. Was such a one-way bias observed in Skyllakou? Does this model configuration address model bias in a way that enhances predictive skill? From my quick read of Skyllakou it appears there is often a positive bias (overprediction) of $PM_{2.5}$ mass concentrations. To what degree does this new model process exacerbate bias and error?*

Skyllakou et al. (2021) found that PMCAMx had a small fractional bias (5%) and fractional error (25%) for the annual average $PM_{2.5}$ concentrations of 1067 measurement stations in the US. Given that the effect of the extension of the model on the total fine PM mass is small (of the order of 1%), this does not result in any noticeable change in its already very good performance for dry fine PM. We do stress now in the revised paper that the major effect of including the SOA water in the simulations is the increase of the predicted aerosol water with implications that include its climatic effects (increase of the aerosol direct effect), visibility, atmospheric chemistry. The corresponding effects of the extension on reproducing the fine PM mass, at least in the area studied, are minor to negligible.

*(13) Starting at line 217: "Aerosol liquid water directly affects the PM sensitivity and dry deposition rates, with direct implications for emissions control policy." It reads awkwardly to introduce these new ideas in the last paragraph of the manuscript.*

We have deleted these sentences and focused the last paragraph on the significance of the SOA water for climate change, visibility and aerosol chemistry.

*(14) The finding that increasing the amount of liquid water increasing nitrate concentrations is an important finding in the spirt and context of this sentence – but the authors gloss over this.*

Thank you for this comment. The effects on predicted nitrate are discussed in the end of the first paragraph of the conclusions. We have extended that discussion and also added this point to the concluding paragraph of the manuscript.

*(15) Table S1: Can the authors provide quantitative meaning or context for "low", "high" and "modest"? How does RH change in these areas?*

We have replaced the qualitative metrics with predicted concentrations and added information about the RH in these areas.

*(16) Sacramento is listed as "low" SOA in Table S1. Sacramento is one of the top 20 most polluted cities in U.S. AMS studies in Davis, CA & Cool, CA (i.e., near Sacramento) are heavily organic dominated. Can the authors defend the choice to characterize Sacramento as 'low'?*

Following the advice of the reviewer (see also our response to Comment 15) we have replaced the qualifiers with quantitative metrics (concentrations). The predicted annual average SOA concentration in Sacramento by PMCAMx is now shown in the table.

**Editorial**

*(17) The months used for the seasonal definitions are not provided.*

We now state clearly the months used for the seasonal definitions (DJF for winter, MAM for spring, JJA for summer, and SON for fall).

*(18) The y-axis in the first row of Fig. S7 is log scale. Why? There should be a note in the Figure caption each time the axes differ.*

We have chosen a log scale here to show clearly both the relatively small average and the large range of high values. This is now noted in the corresponding figure caption.

*(19) The authors rely on some supplemental figures heavily, referring to them many times. They should probably be in the main text.*

We have followed the advice of the reviewer and moved Figures S2, S4 and S7 to the main text.

---

## Author Comment (AC2)

**Responses to the Comments of Reviewer 2**

*(1) With the abstract, I thought the highlight of this manuscript should be that authors have incorporated ISORROPIA-lite into the chemical transport model therefore letting the 3-D models capable of considering impacts of aerosol water associated with organic aerosol (ALW_org) on the partitioning of semi-volatile vapors which excited me for a while because this is really important, however this was already done in Kakavas et al. (2022). Impacts of increased aerosol water on PM$_1$ aerosols including their chemical compositions was also evaluated in Kakavas et al. (2022) although focused region in Kakavas et al. (2022) is Europe.*

Kakavas et al. (2022) focused on the development of ISORROPIA-lite and the comparison of the predictions of a CTM using ISORROPIA-lite vs. ISORROPIA-II. The simulations carried out in Kakavas et al. (2022) was for one late-spring early summer month in Europe – which is very limited in scope, just enough to showcase the new capability of the model, evaluate its computational efficiency and the potential importance of adding organic water uptake. Most of the related conclusions were tentative given that only one warm month was simulated.

The present work thoroughly analyzes organic water uptake impacts over one simulated year (not just one month as done in Kakavas et al., 2022) and in quite a different geographical area (US here versus Europe in Kakavas et al., 2022). There are significant differences, but also similarities in the predicted changes and effects of SOA water. In the revised paper, we first stress the differences between the two studies but then proceed with discussions on their similarities.

*(2) In view of this, this manuscript should advance further the scientific understanding of the significant roles of ALW_org in atmospheric chemistry simulations, however, this manuscript just looks like a report of a sensitivity test of hygroscopicity parameter kappa over United states. The presentations of the results focus only on percentage increase/decrease of PM$_1$ levels and aerosol water in different regions or sites, no insightful analysis was done. Most importantly, the model performance of SOA simulations was not evaluated against observations at all. It was well known that the performance of current chemistry models in simulating the heterogeneous/multi-phase formations of SOA is not satisfactory (Miao et al., 2020) and might significantly underestimate SOA mass concentrations in regions that SOA formations associated with heterogeneous/multi-phase reactions prevail, therefore numbers reported in this study might not be convincing at all.*

We have followed the advice of the reviewer and extended the scientific discussion of the factors that contribute to the increase of the predicted SOA water, both as a fraction of the total but also as absolute concentrations. We also focus more on the specific periods and areas where these increases are predicted.

We do discuss the evaluation of the model predictions for both PM$_{2.5}$ total concentration and composition. Given the small changes on average to these due to the inclusion of the SOA water the PM$_{2.5}$ performances does not change, but it was very good to start with. The performance of PMCAMx regarding annual average OA is actually quite good in these simulations with a fractional bias of 5% and a fractional error of 26% in the 306 stations in the US. For daily average concentrations the performance is also quite encouraging with a fractional bias of 15% and a fractional error of 56%. This information is provided in the revised paper. We do stress now that the major change in the model predictions is on the aerosol water concentrations.

*(3) In addition, as demonstrated by authors, the variations of organic aerosol hygroscopicity (Kappa_OA) was also very important, however, the usage of Kappa_OA were not discussed. In general, discussions of this manuscript are very casual, and literature reviews about significant roles of ALW_org in atmospheric chemistry simulations is poor, for example, previous achievements regarding important roles of ALW_org are not discussed at all (Pye et al., 2017; Jathar et al., 2016; Li et al., 2020).*

We have followed the advice of the reviewer and extended our discussion and choices of the organic aerosol hygroscopicity parameter providing the corresponding references. We have also added a paragraph about the importance of the aerosol water in atmospheric chemistry simulations.

**Specific comments**

*(4) L40-42, The number of literatures with quantitative determination of aerosol water is relatively small, therefore, following references should also de included here: (Bian et al., 2014; Deetz et al., 2018; Kuang et al., 2018; Wu et al., 2018; Gopinath et al., 2022).*

We have added a brief discussion of the previous efforts to measure the atmospheric aerosol water. These include the papers suggested by the reviewer plus a few more from our group.

*(5) L48 Li et al. (2019) should be included here.*

We have included this reference at this point of the manuscript.

*(6) L51 Kuang et al. (2020).*

The reference has been added.

*(7) L59 Secondary aerosol formations.*

We have rephrased this sentence.

**References**

Bian, Y. X., Zhao, C. S., Ma, N., Chen, J., and Xu, W. Y.: A study of aerosol liquid water content based on hygroscopicity measurements at high relative humidity in the North China Plain, Atmos. Chem. Phys., 14, 6417-6426, 10.5194/acp-14-6417-2014, 2014.

Deetz, K., Vogel, H., Haslett, S., Knippertz, P., Coe, H., and Vogel, B.: Aerosol liquid water content in the moist southern West African monsoon layer and its radiative impact, Atmos. Chem. Phys., 18, 14271-14295, 10.5194/acp-18-14271-2018, 2018.

Gopinath, A. K., Raj, S. S., Kommula, S. M., Jose, C., Panda, U., Bishambu, Y., Ojha, N., Ravikrishna, R., Liu, P., and Gunthe, S. S.: Complex Interplay Between Organic and Secondary Inorganic Aerosols With Ambient Relative Humidity Implicates the Aerosol Liquid Water Content Over India During Wintertime, Journal of Geophysical Research: Atmospheres, 127, e2021JD036430, https://doi.org/10.1029/2021JD036430, 2022.

Jathar, S. H., Mahmud, A., Barsanti, K. C., Asher, W. E., Pankow, J. F., and Kleeman, M. J.: Water uptake by organic aerosol and its influence on gas/particle partitioning of secondary organic aerosol in the United States, Atmospheric Environment, 129, 142-154, https://doi.org/10.1016/j.atmosenv.2016.01.001, 2016.

Kuang, Y., Zhao, C. S., Zhao, G., Tao, J. C., Xu, W., Ma, N., and Bian, Y. X.: A novel method for calculating ambient aerosol liquid water content based on measurements of a humidified nephelometer system, Atmospheric Measurement Techniques, 11, 2967-2982, 10.5194/amt-11-2967-2018, 2018.

Kuang, Y., Xu, W., Tao, J., Ma, N., Zhao, C., and Shao, M.: A review on laboratory studies and field measurements of atmospheric organic aerosol hygroscopicity and its parameterization based on oxidation levels, Current Pollution Reports, 10.1007/s40726-020-00164-2, 2020.

Li, J., Zhang, H., Ying, Q., Wu, Z., Zhang, Y., Wang, X., Li, X., Sun, Y., Hu, M., Zhang, Y., and Hu, J.: Impacts of water partitioning and polarity of organic compounds on secondary organic aerosol over Eastern China, Atmos. Chem. Phys. Discuss., 2020, 1-35, 10.5194/acp-2019-1200, 2020.

Li, X., Song, S., Zhou, W., Hao, J., Worsnop, D. R., and Jiang, J.: Interactions between aerosol organic components and liquid water content during haze episodes in Beijing, Atmos. Chem. Phys., 19, 12163-12174, 10.5194/acp-19-12163-2019, 2019.

Miao, R., Chen, Q., Zheng, Y., Cheng, X., Sun, Y., Palmer, P. I., Shrivastava, M., Guo, J., Zhang, Q., Liu, Y., Tan, Z., Ma, X., Chen, S., Zeng, L., Lu, K., and Zhang, Y.: Model bias in simulating major chemical components of PM2.5 in China, Atmos. Chem. Phys., 20, 12265-12284, 10.5194/acp-20-12265-2020, 2020.

Pye, H. O. T., Murphy, B. N., Xu, L., Ng, N. L., Carlton, A. G., Guo, H., Weber, R., Vasilakos, P., Appel, K. W., Budisulistiorini, S. H., Surratt, J. D., Nenes, A., Hu, W., Jimenez, J. L., Isaacman-VanWertz, G., Misztal, P. K., and Goldstein, A. H.: On the implications of aerosol liquid water and phase separation for organic aerosol mass, Atmos. Chem. Phys., 17, 343-369, 10.5194/acp-17-343-2017, 2017.

Wu, Z., Wang, Y., Tan, T., Zhu, Y., Li, M., Shang, D., Wang, H., Lu, K., Guo, S., Zeng, L., and Zhang, Y.: Aerosol liquid water driven by anthropogenic inorganic salts: Implying its key role in haze formation over the North China Plain, Environmental Science & Technology Letters, 10.1021/acs.estlett.8b00021, 2018.

---

## Author Response (AR2)

**Responses to the Comments of the Reviewers**

**Reviewer 3**

**(1)** This study incorporated aerosol water associated with secondary organic aerosol (SOAW) in a thermodynamic model and accounted for the influence of SOAW on the partitioning of inorganic species like sulfate, nitrate and ammonium. The updated thermodynamic model (ISORROPIA-lite) was integrated with a chemical transport model (PMCAMx) to evaluate the impact of SOAW on particulate matter (PM) concentration over the US. It was found that while SOAW significantly increased total aerosol water by 20-50%, the corresponding enhancement in PM mass concentration was mostly negligible (around or less than 2%). The study concluded that it was important to account for the influence of SOAW on PM due to changes in particle size and thus optical properties and deposition rate. Major revisions are suggested for the manuscript before acceptance for publication.

Our responses and corresponding changes to the manuscript (in blue) follow each comment of the reviewer (in black).

**(2)** This study was directed at addressing an important gap in the current modeling of PM by linking the thermodynamics of inorganic species and SOA. However, the manuscript missed key discussions that would serve as the basis for this study. Specifically, the manuscript did not explain the current inorganic aerosol formation pathways in ISORROPIA-lite and how much these pathways are expected to be influenced by enhanced aerosol water from SOA. Some novel aqueous pathways for inorganic aerosol formation and synergistic effects between different inorganic components have been found in recent studies (e.g., aqueous conversion of $SO_2$ to SO4 by dissolved NO2 and NH3,[1] hydroxymethansulfonate formation,[2] air-water interface catalyzed reactions,[3] black carbon catalyzed reactions[4] and etc). Therefore, it is important to describe what aqueous inorganic pathways ISORROPIA-lite accounts for, whether the novel pathways are relevant for the simulation over the US, and what the limitations are if some potentially important pathways are not accounted for. For now, the model sensitivity in PM mass to SOAW is not presented in context of these discussions.

We now clarify in the revised manuscript that ISORROPIA-lite is an aerosol thermodynamics model, therefore it focuses on the simulation of the thermodynamic equilibrium of atmospheric aerosol. Processes like the one mentioned by the reviewer (bulk aqueous-phase reactions, heterogeneous interface reactions, catalyzed reactions, etc.) are usually described in the corresponding chemical transport module (aqueous-phase chemistry, inorganic and organic aerosol chemistry, etc.) and not in the aerosol thermodynamics module. A brief description of the processes described by PMCAMx currently (e.g., the aqueous oxidation of $SO_2$ to sulfate by $NO_2$, the formation of hydroxymethansulfonate, etc.) has been added in the PMCAMx description section.

**(3)** Other suggestions include to discuss the potential impact of increased aerosol water on the radiative properties of PM over the US. Specifically, what is the change in direct radiative forcing due to more scattering caused by increased aerosol water?

A brief discussion of the expected magnitude of these effects has been added to the revised paper.

**References**

(1) Peng, J.; Hu, M.; Guo, S.; Du, Z.; Zheng, J.; Shang, D.; Levy Zamora, M.; Zeng, L.; Shao, M.; Wu, Y.-S.; Zheng, J.; Wang, Y.; Glen, C. R.; Collins, D. R.; Molina, M. J.; Zhang, R. Markedly Enhanced Absorption and Direct Radiative Forcing of Black Carbon under Polluted Urban Environments. Proc. Natl. Acad. Sci. U.S.A. 2016, 113 (16), 4266–4271. https://doi.org/10.1073/pnas.1602310113.

(2) Moch, J. M.; Dovrou, E.; Mickley, L. J.; Keutsch, F. N.; Cheng, Y.; Jacob, D. J.; Jiang, J.; Li, M.; Munger, J. W.; Qiao, X.; Zhang, Q. Contribution of Hydroxymethane Sulfonate to Ambient Particulate Matter: A Potential Explanation for High Particulate Sulfur During Severe Winter Haze in Beijing. Geophys. Res. Lett. 2018, 45 (21). https://doi.org/10.1029/2018GL079309.

(3) Hung, H.-M.; Hoffmann, M. R. Oxidation of Gas-Phase SO 2 on the Surfaces of Acidic Microdroplets: Implications for Sulfate and Sulfate Radical Anion Formation in the Atmospheric Liquid Phase. Environ. Sci. Technol. 2015, 49, 13768–13776. https://doi.org/10.1021/acs.est.5b01658.

(4) Zhang, F.; Wang, Y.; Peng, J.; Chen, L.; Sun, Y.; Duan, L.; Ge, X.; Li, Y.; Zhao, J.; Liu, C.; Zhang, X.; Zhang, G.; Pan, Y.; Wang, Y.; Zhang, A. L.; Ji, Y.; Wang, G.; Hu, M.; Molina, M. J.; Zhang, R. An Unexpected Catalyst Dominates Formation and Radiative Forcing of Regional Haze. Proc. Natl. Acad. Sci. U.S.A. 2020, 117 (8), 3960–3966. https://doi.org/10.1073/pnas.1919343117.

**Reviewer 4**

**(1)** The manuscript simulated the SOAW and investigated its effects on inorganic species of PM1 over the US. It is revealed that SOAW can increase the average aerosol water by a factor of up to 2 and generally increase the partitioning of ammonium nitrate. This is an interesting topic and has important implications for understanding the role of the environmental effects of SOA. The manuscript is clearly written and ACP is an appropriate venue, but I do have some concerns regarding methods and data analysis that must be addressed before the paper can be considered for publication.

Our responses and corresponding changes to the manuscript (in blue) follow each comment of the reviewer (in black).

**(2)** The effects of SOAW on the partitioning of organic species were not considered. How would SOAW impact the partitioning of organic vapors? More importantly, what is the role of SOAW in aqueous phase chemistry?

ISORROPIA-lite is an aerosol thermodynamics model, therefore it focuses on the simulation of the thermodynamic equilibrium of atmospheric aerosol. For the simulation of organic aerosol the volatility basis set (VBS) approach is used. Therefore, partitioning of organic vapors is described in the corresponding chemical transport module and not in the inorganic aerosol thermodynamics module. A brief description of the organics species partitioning in PMCAMx has been added in the PMCAMx description section.

**(3)** The fractional bias of the total PM$_{2.5}$ and its major components of the PMCAMx was typically 30%. This is significantly higher than the 1% increase in PM levels induced by SOAW. The 1% difference is so small (within the uncertainty of the PM simulation) that it is not sound to compare the SOAW effects on the absolute changes of PM components across different regions. The authors should focus more on the fractional changes of PM induced by SOAW instead of the absolute changes. That said, for example, Figure 6 can be removed or go to the supplemental information.

Please note that the fractional bias was a lot lower. For example, it was 5% for the annual average PM$_{2.5}$ concentrations in more than 1000 measurement stations. The bias was 5% for OA, 17% for sulfate, 6% for EC, etc. The highest bias was for nitrate. We do believe that the absolute changes are important (e.g., for the estimation of the direct effect) as are the fractional changes suggested by the reviewer. In the revised paper we present both in most cases. We have followed the advice of the reviewer and moved Figure 6 to the Supplemental Information.

**Specific comments**

**(4)** Title: The paper discussed the effects of SOAW on PM1, not fine PM. Please clarify.
We have added the PM$_1$ to the title.

**(5)** Lines 92-94: It would be helpful to elaborate on the chosen values of SOA density and κ here.

A brief discussion and the corresponding references have been added.

**(6)** Lines 110-114: The four volatility bins of SOA only cover the SVOC range. Why were these bins selected for the simulation? LVOC should also be considered.

The LVOCs (and the ELVOCs) are implicitly included in the lowest volatility bin of this version of the VBS used in PMCAMx. These compounds are always in the particulate phase in these simulations and therefore the addition of lower bins increases the computational cost without changing the predicted organic aerosol concentration. A brief explanation of this choice has been added to the revised paper.

**(7)** Lines 122-124: This is confusing. Why were the model predictions more reliable for RH values above 95%?

We have rephrased this rather confusing statement.

**(8)** Lines 129-132: Why was κ=0.2 selected as the maximum value, not 0.3 given that the κ of SOA is 0.1-0.3?

The κ=0.3 represents an extreme case for the simulation of the average effects over the US. The two current simulations also provide a good estimate of the corresponding uncertainty. A brief explanation for our choice has been added to the paper.